# Experimental Investigations of Impact Damage Influence on Behavior of Thin-Walled Composite Beam Subjected to Pure Bending

**DOI:** 10.3390/ma12071127

**Published:** 2019-04-06

**Authors:** Tomasz Kubiak, Lukasz Borkowski, Nina Wiacek

**Affiliations:** Department of Strength of Materials, Lodz University of Technology, Stefanowskiego 1/15, 90-924 Lodz, Poland; lukasz.borkowski@p.lodz.pl (L.B.); ninawiacek@gmail.com (N.W.)

**Keywords:** thin-walled laminates, impact pre-damage, channel section beam, postbuckling, failure

## Abstract

The paper deals with buckling, postbuckling, and failure of pre-damaged channel section beam subjected to pure bending. The channel section beams made of eight-layered GFRP laminate with different symmetrical layups have been considered. The specimens with initially pre-damaged web or flange were investigated to access the influence of impact damage on work of thin-walled structure in the full range of load till failure. The bending tests of initially pre-damage beams have been performed on a universal tensile machine with especially designed grips. The digital image correlation system allowing to follow the beam deflection have been employed. The experimentally obtained results are presented in graphs presenting load-deflection or load vs. angle of rotation relations and in photos presenting impact damages areas before and after bending test. The results show that the impact pre-damages have no significant influence on the work of channel section beams.

## 1. Introduction

Modern composites are light and durable, their properties can be adjusted by fitting them to the shape and manner of loading the designed structure. The disadvantage of laminates based on fiber-reinforced resins is their manufacturing cost, rather fragile behavior, the inability to work at high temperatures, unknown behavior after minor operational damage. Despite that, a different type of laminates (i.e., FRP, FML) manufactured in a different way (i.e., autoclaving or pultruded method) and different types of sandwich or FGM structures are widely used. They are more and more popular in many branches of industry, i.e., aircraft industry, aerospace engineering, automotive industry or civil engineering and many others. 

Nowadays, in worldwide literature it is quite easy to find papers dealing with thin-walled laminates—structures made of composites. Taking into account such structures, the following two aspects should be mentioned: (i) buckling, postbuckling, and failure of thin-walled structures, (ii) not obvious work and failure of composites. As it is well known, buckling and postbuckling behavior of thin-walled structures are very important, even more than their strength. The same range of importance for thin-walled composite (especially laminate) structures seems to be initial damages i.e., operating damages (barely visible impact damage). 

The number of papers devoted to buckling and postbuckling behavior of thin-walled structures made of composite materials in the latest years has grown rapidly, but in author’s opinion they still need further investigations. It is well-known that papers dealing with above-mentioned issue started to appear in the 1990s. Modern composite beams are made using an autoclaving or pultrusion method. The pultruded beams are generally used in civil engineering. For example, a paper written by Ascione, Feo, and Mancusi [1,2,3], analyzes the deflection and pre-buckling behavior of such structures. They developed the new method based on a 1D beam model for FEM calculations and compared the obtained results with those obtained based on the Vlasov theory or the Timoshenko beam theory. 

On the contrary, the structures (also beams) used in aerospace structures [4], aircraft wing girders [5,6], blades of helicopters [7] or girders of wind turbine blades [8], are made in an autoclaving process. In world wise literature, the papers showing the results of experimental investigations on thin-walled structures made of composite materials can also be easily found, for instance [9,10,11,12,13]. Most of them present a comparison of the results of experimental investigations with the results obtained in FEM simulations. 

Considering the thin-walled structures made of laminate and possibility of minor damage during operation, it has been decided to check the influence of operating damages on thin-walled laminate structures in the whole range of load. Generally, the influence of impact damage on the composite structure resistance is developed based on standard tests CAI [14,15]. It should be notated, that it is possible to find a significant number of papers devoted to low-velocity impacts (LVIs) as well as to CAI [16], where the typical size and shape specimen (i.e., a plate with overall dimensions 150 mm × 100 mm) have been used. Nevertheless, the influence of different geometry and load parameters on the type and area of occurring damages have been also investigated. The following exemplary paper could be mentioned: (i) the influence of impact position on damages have been investigated by Sun and Hallett [17], (ii) the effects of ply clustering on polymer-based laminated composite plates subjected to impact load have been considered by Gonzales et al. [18], (iii) the influence of layer arrangement, impact energy, and ball velocity on damage area, strain in time relation, delamination area vs. impact energy in the case of single impact and multiple impacts based on not standard specimen dimensions (i.e., panels 900 mm × 1100 mm) were tested by Nassr et al. [19], (iv) the influence of impact energy and impact force on delamination area of GFRP plates with rectangular and circular shape have been investigated by Cantwell [20]. (v) Further, the influence of drop off configuration on impact load response have been analyzed by Abdulhamid et al. [21], (vi) Shyr and Pan [22] showed multi-elements analysis of impact resistance of fiber glass composite materials, (vii) the tensile tests on initially pre-damaged thin-ply angle-ply pseudo-ductile carbon fiber laminates have been investigated by Prato et al. [23], (viii) Gliszczynski et al. [24,25] investigated the low velocity impact damages in plate and channel section profile on size and shape of damage with comparison to standard CAI tests. Additionally, the repaired and non-repaired specimens after impact was tested by Kumari et al. [26] in compression test, who investigated their behaviour till damage.

Concluding the above, it can be said that it is quite easy to find results of investigations dealing with buckling and postbuckling behavior as well as showing the influence of impact pre-damage on the resistance of laminates, but in world literature there are not enough papers dealing with influence on impact pre-damages on buckling, postbuckling and failure of thin-walled laminates beam or columns. However, it should be mentioned that similar investigation, but on steel thin-walled tubes with rectangular cross-section, has been performed by Chen et al. [27], who checked the influence of lateral impact on the axial bearing capacity. The different impact energy, loading position, and width/thickness ratios have been considered to access their influence on the residual axial bearing capacity the failure mode, initial stiffness and ductility of tested specimens. 

Additionally, it is worth to notice that earlier published papers deal with the influence of impact pre-damages on buckling and postbuckling behavior of columns subjected to uniform compression [28,29]. This paper presents the results of experimental investigation of pre-damaged by lateral low-velocity impact (barely visible impact damage) channel section columns subjected to pure bending. Obtained results allow to access the influence of initial impact damage on buckling load, postbuckling behavior, and failure load. 

In author’s opinion, the knowledge of thin-walled laminate beam behavior in the whole range of load looks to be necessary for designers involved in preparing standards to design the lighter, more durable and safe structures. The above will be possible if more results for structures with operating damages (impact damages) and their influence on postbuckling and failure are presented.

## 2. Object of Investigations 

The analysis of the influence of impact pre-damage on buckling, postbuckling, and failure of thin-walled beam was carried out. As an object of analysis, the thin-walled channel section beams (Figure 1) with cross-section dimension of a × b × t = 84 mm × 40 mm × 2.1 mm (a—a web width, b—a flange width and t—a wall thickness) were considered. The beams for tests were produced using an autoclaving technique to ensure the best quality so that possible production imperfections (inclusions, voids) did not affect the test result. The unidirectional glass fiber reinforced polymer (GFRP) pre-impregnate produced by GURIT have been chosen. The eight layers laminate with following lay-ups were produced and then tested: [45/−45/90/0]_s_ further denoted as Cb1, [90/−45/45/0]_s_—Cb2, [0/−45/45/90]_s_—Cb3, [0/90/0/90]_s_—Cb4 and [45/−45/45/−45]_s_—Cb5. The material properties (E_1_, E_2_—Young moduli in fiber direction and transverse to fiber direction, respectively, G_12_—Kirchhoff modulus, ν_12_—Poisson ratio, ultimate stresses: in fiber direction in case of tension X^T^ and compression X^C^, in transverse to fiber direction in case of tension Y^T^ and compression Y^C^, S—ultimate shear stress) of unidirectional plies are presented in Table 1 [30].

Pre-damaged channel section beams were taken for experimental tests. The low-velocity impact damage (pre-damage) was introduced in mid-span of tested beam and two different places, i.e., mid-width of the web and mid-width of the flange. Thus, the determined series of data were denoted in the following form Cbx-W or Cbx-F where x refers to the analyzed arrangement of layers and F or W concerns the considered element (F—flange, W—web) of the channel section profile. 

The impact damages were introduced with the energy of impact equal to 20 J using an impactor with the hemispherical shape of the tip. A detailed description of the stand to introduce the initial damage can be found in the authors’ previous papers [24,25,28]—only for clarification, the schemes of the test stands for impact damage introduced in web and flange are presented in Figure 2.

## 3. Experimental Test Stand

The initially damaged beams were subjected to pure bending in the plane of the flange (plane bending due to the symmetry of cross-section). The four-point bending test was performed employing the test stand [30]. The dimensions describing the span of considered beam and the span of the load are presented in the scheme of the test stand (Figure 3).

The bending after impact tests of the considered profile were conducted on a universal testing machine (UTM) produced by Instron (Norwood, MA, USA) and modernized by Zwick-Roel (Ulm, Germany). The tests were performed at a constant velocity of the cross-bar equal to 1.5 mm/min. The loading force and the displacement in the points where the load was applied were obtained directly from the machine sensors. To determine the deflection of the beam in the whole range of load, a digital image correlation system Aramis produced by GOM (Braunschweig, Germany) was employed. The data from the DIC were captured with a frequency of 1 Hz together with load transferred directly from the testing machine. The whole test stand is presented in Figure 4.

The collected data allow to determine bending moment vs. angle of rotation or bending moment vs. deflection plots. The obtained results allow to determine buckling loads and modes, analyze postbuckling behaviors and determine the failure loads with corresponding failure mechanisms.

## 4. Results of Experimental Tests

The results of the experimental tests are presented and discussed in this section. They are presented in the form of (i) pictures showing impact damage before and after bending test, buckling mode and failure mechanism, (ii) graphs presenting the relation between load and deflection or load vs. angle of rotations. 

Impact pre-damages for all considered cases of layups and place of impact are presented in Table 2. In most cases that were analyzed, the size of damage, as well as the character of damage, do not change after bending tests, which have been performed till failure of the beam. 

It also should be mentioned that damage due to the bending load have not been started at place of the initial pre-damages Even in the cases denoted as Cb3-F and Cb1-W (see Table 2) the impact damage is very close to damages due to the bending load, and the impact pre-damage has not initiated the failure in the bending test. It should be noted that in case Cb2-F (cf. Figure 5) the failure due to bending goes through the initial impact pre-damage.

Concluding all test observations of which results are presented in Table 2 and Figure 5, it can be said that failure due to bending starts from matrix and fiber cracking on the flange-web edge (c.f. Figure 5c) and then propagates along the flange to the free edge and along the web. The direction of damage propagation corresponds to layer arrangements.

For cases denoted as Cb5 where the fibers in relation to the axis of the beam longitudinal edges are located at an angle of ±45, the plastic behavior is observed—the beams did not break and after unloading permanent deflection was visible (see Figure 6).

The beam’s behavior under pure bending till failure can be analyzed on the basis of the bending moment vs. angle of rotations at support curves (Figure 7) as well as the bending moment—relative deflection curves (Figure 8). The relative deflection is calculated in chosen cross-section as a difference between mid-with of the web deflection and web-flange edge deflection. Observing the course of the curves in Figure 7 and Figure 8, it can be said that the place of initial pre-damage has no significant influence on beam behavior, which is also proved by the further presented results.

Analyzing the course of load vs. angle of rotation curves (Figure 7) it is visible that beams with the most cases of layer arrangement have the brittle behavior—almost straight line till failure load. Only in the case denoted as Cb5 the plastic behavior is observed—the bending moment vs. angle of rotation at support have a nonlinear relation, which was proved by permanent deflection after unloading, visible in Figure 6. Such behavior can also be observed on load-relative deflection curve (Figure 8e), where for high bending moment (>450 Nm) the relative deflections are constant, which means that there is no relation between the deflection of mid-width of the web and web-flange edge in the chosen cross-section.

The results of the experimental test also allow to determine the buckling load (Table 3) with a corresponding buckling mode (Table 4).

It is well known that especially in the case of experimental tests, the structure subjected to bending the bifurcation load does not exist. A different approximation method like for example “top of the knee”, P-w^2^ [31] or proposed by Teter and Kolakowski [32] allows to estimate the buckling load. The estimated buckling loads based on load vs. relative deflection curves (Figure 8)—M_b1_ after employing “top of the knee” method [31] and on the basis of load vs. angle of rotation curves (Figure 7)—M_b2_ employing Teter–Kolakowski method [32] are presented in Table 4. Taking into account the similarity of load–deflection and load–angle of rotations curves, as well as the buckling modes for given layer arrangement and both places of pre-damages, it can be noted that better results of the buckling load have been obtained based on load—angle of rotation curves [32]—the results are closer to each other.

As it can be observed on pictures presented in Table 4, the buckling modes differ only due to layer arrangement, the places of impact pre-damage have no significant influence on the buckling mode as well as the buckling load. Only in the case Cb4, the change of buckling mode depends on the place of impact pre-damage—the impact introduced on the web causes the creation of the initial deflection, which translates into the deflection corresponding to the buckling half-wave.

The failure loads for the cases from Cb1 to Cb4 and maximum load collected during the test for case Cb5 are presented in Table 5. Due to no damage and plastic behavior of beam with layer arrangement denoted as Cb5, only the maximal load with the corresponding angle of rotation at support are presented—those results cannot be compared with others. It only can be said that such layer arrangement causes the beam to be very flexible—the beam subjected to similar or lower bending moment which in other cases leads to damage in this one leads to rotate at the support by more than twice the angle measured in other cases.

Comparing the failure bending moments for beams with pre-damaged webs and flange with no influence of place of pre-damage on the failure due to the bending moment, the results listed in Table 5 appear surprising. In relation to the previously described results, they should have similar values, however, beams with pre-damaged flanges carry a larger bending moment than those with a pre-damaged web. The differences are from approx. 3–5% for beams Cb1, Cb2 and Cb4 to c.a. 20% for case denoted as Cb2. After all, slightly higher load-bearing values for beams with pre-damaged flanges than beams with pre-damaged webs should be considered as an accident. Differences in load capacities of up to 5% can be considered with a high probability of spreading associated with the measurement accuracy and accuracy of beam manufacturing. The 20% difference for the Cb2 case is simply due to the place where the damage due to bending occurred—it is not related to the place of initial damage in any way. The Cb2-W beam was destroyed in the place where the beam was fixed in the grip, and the Cb2-F beam in the mid-span (see Table 2). Therefore, the lower failure bending moment of 732 Nm for beam with pre-damaged web in comparison to failure bending moment equal to 890 Nm for beam with pre-damaged web could be a coincidental result of the way of mounting the beam in the aluminum grip (Figure 3b). The above results show no differences in load-deflection and load-angle of beam rotation curves as well as no correlation between pre-damages and failure due to bending.

It is worth to mention that observations based on obtained results are similar to the other authors’ investigations performed on columns subjected to uniform compression [28,29].

## 5. Conclusions

In this paper, buckling, postbuckling and failure of pre-damaged channel section beam subjected to pure bending have been investigated. The channel section beams made of 8-layered GFRP laminate with two different symmetrical layups was used. The specimens with impact pre-damages on web or flange have been investigated to access the influence of initial impact damage on work of a thin-walled structure in the full range of load till failure. 

Analyzing the results obtained from experimental research, it can be concluded that introduced impact damages do not have an influence on the buckling load as well as the failure load of channel section beams under consideration. Analysis of the influence of layer arrangement on sensitivity to size and shape of impact damages, and then on the operation of a pre-damaged beam subjected to pure bending also show that such relations have not been observed. However, in case of beams with layups where the outer layers have fiber inclined to the edges at an angle of
45° (Cb-1 and Cb-5) the size of delamination after impact in flange is higher than in other cases. Despite the above, for these cases, a decrease of buckling or failure loads has not been observed.

It is worth to mention that the introduced damages were small in comparison to channel web or flange dimensions (i.e., their length and width), what could be the main reason of the observed behavior (no influence of initial impact damage). The reason for this can be probably not only the size of pre-damages but also the places of damage introduction are not sensitive enough to affect the beam behavior. 

The obtained results for the considered cases suggest performing further investigations, in which the stiffest part of a beam (i.e., web–flange junction) will be initially damaged—then the influence of pre-damage on beam behavior could be visible.

## Figures and Tables

**Figure 1 materials-12-01127-f001:**
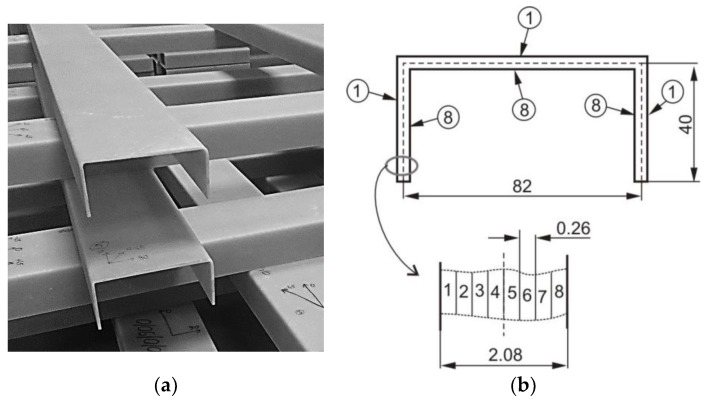
Considered channel section profiles (**a**) with cross-section dimension (**b**).

**Figure 2 materials-12-01127-f002:**
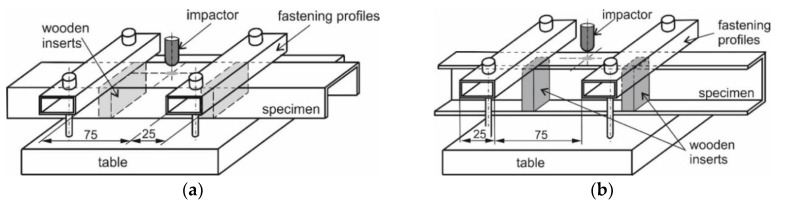
Schemes of profile support in case of impact into web (**a**) and flange (**b**).

**Figure 3 materials-12-01127-f003:**
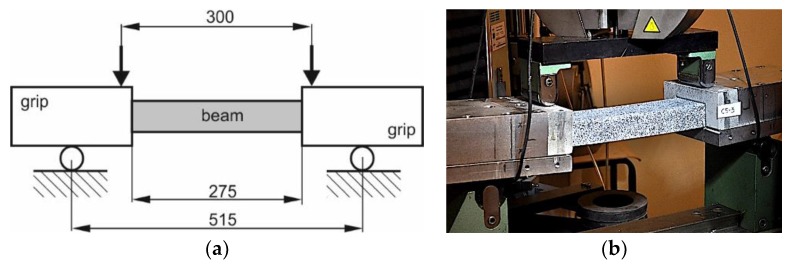
Scheme (**a**) and real test stand (**b**) for bending test.

**Figure 4 materials-12-01127-f004:**
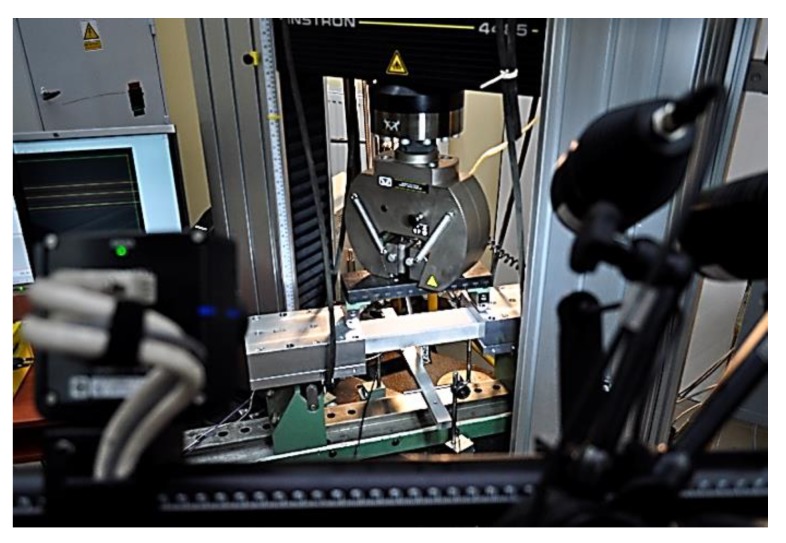
Test stand for four-point bending test.

**Figure 5 materials-12-01127-f005:**
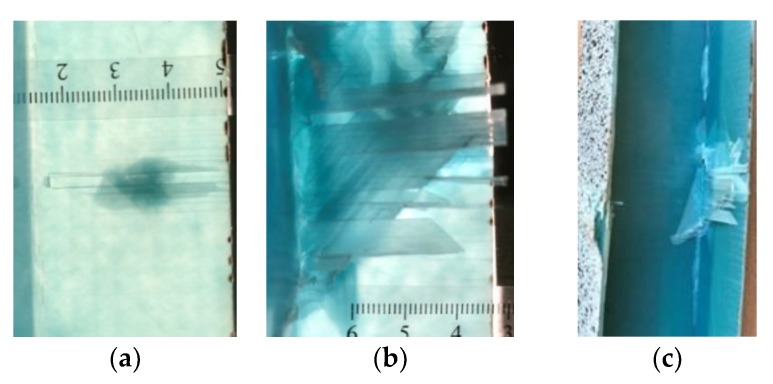
Impact damage before (**a**) and after (**b**) bending test, (**c**) for beam Cb2-F.

**Figure 6 materials-12-01127-f006:**
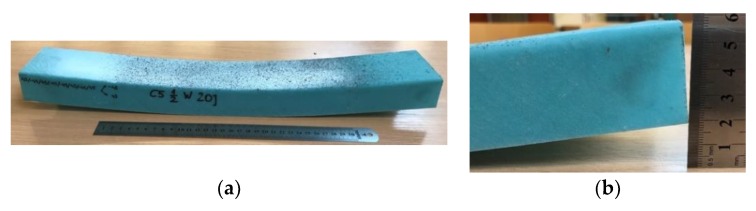
Permanent (**a**) “plastic” deflection in unloaded Cb5 beam with zoom view (**b**).

**Figure 7 materials-12-01127-f007:**
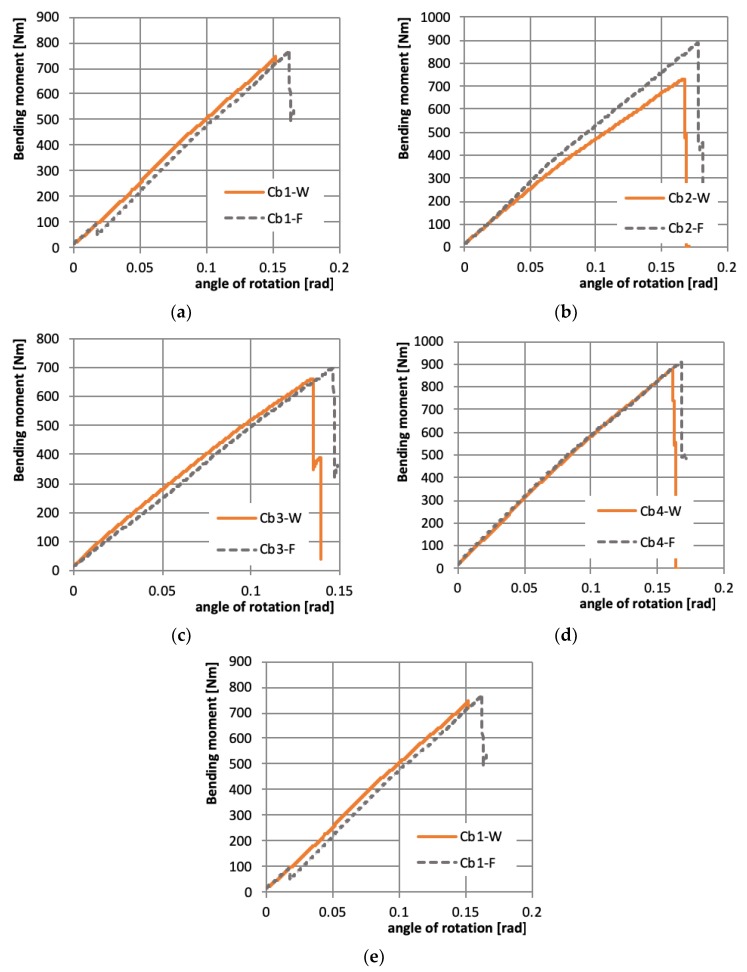
Bending moment vs. angle of rotation at support for pre-damaged beams with different layer arrangements. (**a**) beams with layups Cb1; (**b**) Cb2; (**c**) Cb3; (**d**) Cb4; (**e**) Cb5.

**Figure 8 materials-12-01127-f008:**
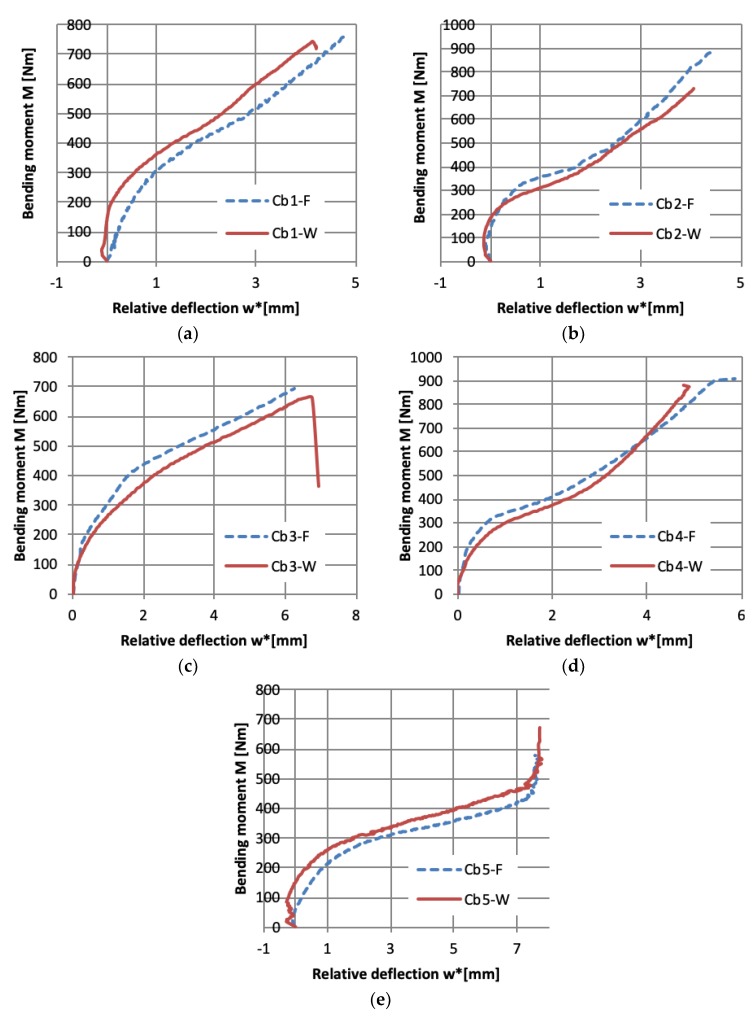
Bending moment vs. relative deflection for pre-damaged beams with different layer arrangements. (**a**) beams with layups Cb1; (**b**) Cb2; (**c**) Cb3; (**d**) Cb4; (**e**) Cb5.

**Table 1 materials-12-01127-t001:** The material properties laminate under consideration [30].

E_1_ (GPa)	E_2_ (GPa)	G_12_ (GPa)	ν_12_ (-)	X^T^ (MPa)	Y^T^ (MPa)	X^C^ (MPa)	Y^C^ (MPa)	S (MPa)
38.5	8.1	2.0	0.27	792	39	679	71	108

**Table 2 materials-12-01127-t002:** Impact pre-damages before and after bending test.

	Flange Pre-Damaged		Web Pre-Damaged	
Layups	Before Bending	After Bending Test	Before Bending	After Bending Test
Cb1	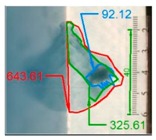	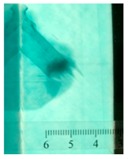	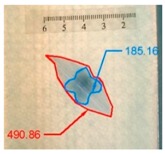	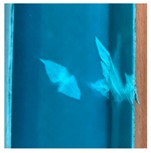
Cb2	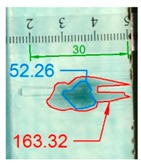	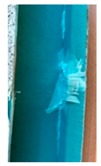	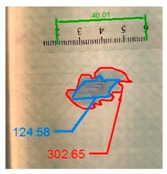	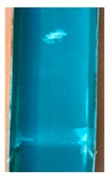
Cb3	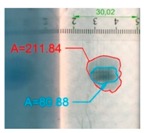	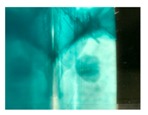	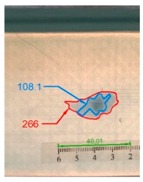	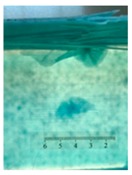
Cb4	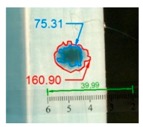	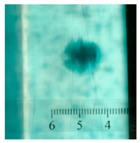	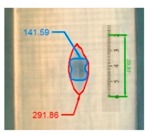	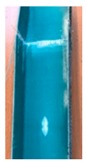
Cb5	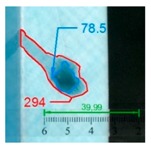	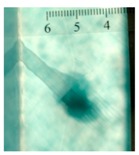	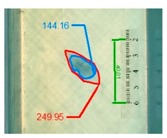	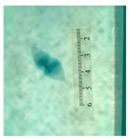

**Table 3 materials-12-01127-t003:** Buckling bending moments.

	Buckling Moment [Nm] M_b1_/M_b2_	
Lay-Up	Web Pre-Damaged	Flange Pre-Damaged
Cb1	320/412	420/426
Cb2	324/332	359/322
Cb3	115/150	185/156
Cb4	240/340	265/325
Cb5	188/175	234/190

**Table 4 materials-12-01127-t004:** Deflection corresponding to buckling mode.

Lay-Up	Web Pre-Damaged	Flange Pre-Damaged
Cb1	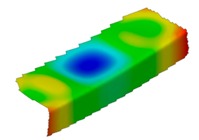	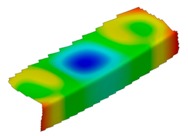
Cb2	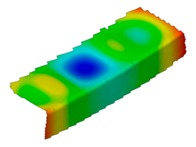	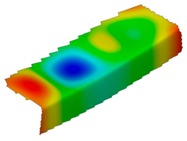
Cb3	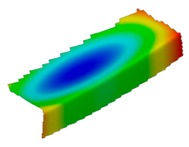	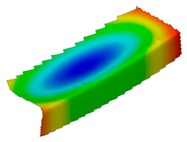
Cb4	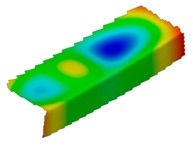	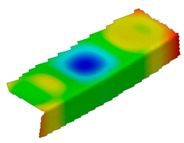
Cb5	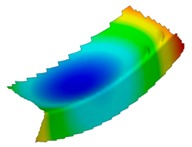	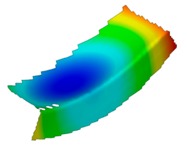

**Table 5 materials-12-01127-t005:** Failure moments with corresponding angle of rotation.

	Failure Moment M_bf_ (Nm)/Corresponding Angle of Rotation (deg)	
Lay-Up	Web Pre-Damaged	Flange Pre-Damaged
Cb1	745/8.7	771/9.2
Cb2	732/9.5	890/10.1
Cb3	663/7.7	699/8.3
Cb4	881/9.6	910/10.0
Cb5	741/21.7 ^1^	582/19.9 ^1^

^1^ No damage, extremely high deflection (plastic deflection c.f. Figure 6).

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
