# Peer review of "Experimental Investigations of Impact Damage Influence on Behavior of Thin-Walled Composite Beam Subjected to Pure Bending"

_materials, 2019, doi:10.3390/ma12071127_

Reviewer 1 Report

Review for materials- 478754

Experimental investigations of impact damage influence on behavior of thin-walled composite beam subjected to pure bending

The authors address an interesting research topic for the journal Materials. It is a well-organized paper but some recommendations should be considered:

1.     The abstract should include a last paragraph indicating a brief resume of results, thereby doing more attractive the paper to the potential readers.

2.     Line 233-236. “Analyzing the results….as failure load”. This sentence is not clear.

3.     In my opinion, the hard and exhaustive work developed in the paper is not reflected in the conclusion section. Thus, authors should make an effort to improve the quality of the conclusions.

4.     From my point of view, last paragraph in Conclusion section (lines 238-239) is not adequate for this point. Please, move such a sentence to the previous section.

5.     Although the number and the selection of references is adequate, it would be advisable to include some papers from the journals of MDPI editorial (Materials, Metals, etc.) related to the topic of the manuscript.

Minor changes:

6.     Please, complete the affiliation of authors according to the journal format.

7.     Line 106: “...denoted as Cb1and…” => Please, include a blank space between “Cb1” and “and”

8.     Line 147: Table 2, instead of Table 1.

9.     Line 184: “(Fig.8)” => Please, include a blank space between “Fig.” and “8”

10.  Please, revise the text size in the paragraph located on lines 212-226.

Author Response

First, we would like to thank the reviewers for their effort and especially for their useful advice related to our paper. We hope that the reviewers will be satisfied with the revisions we have made in our article.

All Reviewer's suggestions have been considered in improved manuscript, i.e.: 

1) brief resume of results has been added to the abstract;

2) unclear sentence in conclusion “Analyzing the results….as failure load” have been corrected;

3) the conclusions have been extended;

4) the last paragraph in Conclusion section have been move to the previous section;

5) the papers related to the topic of the manuscript, which were published in the journals of MDPI editorial (Materials, Metals, etc.) have been add as references in Introduction.

All minor changes suggested by Reviewer have also been done.

Reviewer 2 Report

The paper is interesting and highlights the effect of damage on the flexural behavior of the thin walled composite beam. The experimental research underlying the work is laborious and ample.
However, there are certain aspects that can be improved
:

1. In introduction, paragraphs 20 ... 28 may be missing given that the literature review is achieved through the rest of the introduction.

2. Object of investigations

     2.1. I suggest to explain the notations in the Table. 1.
     2.2. In paragraphs 105-107, two types of composite beams, denoted with Cb1 and Cb2, were mentioned for investigation, in accordance to the layout of the layers. However, in the chap. 4 Results of Experimental Test, five types of samples are presented. What are the plies arrangement of the Cb3, Cb4, Cb5 samples? Should be explicitly mentioned in Chapter 2.

3. I suggest you improve the figures in table 1.

4. In Figures 7 and 8 will suggest to highlight curves with different line types.

Author Response

First, we would like to thank the reviewers for their effort and especially for their useful advice related to our paper. We hope that the reviewers will be satisfied with the revisions we have made in our article.

All Reviewer's suggestions have been considered in improved manuscript, i.e.: 

1) the first part of introduction (lines 20 ... 28) have been remove;

2) the notations used in Table 1 have been explained as well as all plies arrangements - information about layer arrangement for cases deonoted as Cb3, Cb4, Cb5 have been add;

3) authors have tried to improve the poctures presented in Table 2.

4) the curves presented in Figs. 7 and 8 have been highlight.

Round  2

Reviewer 1 Report

Accept in present form